# Design of Oligonucleotide Carriers: Importance of Polyamine Chain Length

**DOI:** 10.3390/polym10121297

**Published:** 2018-11-23

**Authors:** Vadim V. Annenkov, Uma Maheswari Krishnan, Viktor A. Pal’shin, Stanislav N. Zelinskiy, Gayathri Kandasamy, Elena N. Danilovtseva

**Affiliations:** 1Limnological Institute of the Siberian Branch of the Russian Academy of Sciences, 3, Ulan-Batorskaya St., P.O. Box 278, Irkutsk 664033, Russia; acrom@mail.ru (V.A.P.); jt1233@mail.ru (S.N.Z.); danilovtseva@yahoo.com (E.N.D.); 2Centre for Nanotechnology & Advanced Biomaterials (CeNTAB), School of Chemical and Biotechnology, SASTRA University, Thanjavur 613401, Tamil Nadu, India; umakrishnan@sastra.edu (U.M.K.); gayathri.ks86@gmail.com (G.K.)

**Keywords:** polymeric amines, oligonucleotides, critical length, grafted polyamines, gene delivery

## Abstract

Amine containing polymers are extensively studied as special carriers for short-chain RNA (13–25 nucleotides), which are applied as gene silencing agents in gene therapy of various diseases including cancer. Elaboration of the oligonucleotide carriers requires knowledge about peculiarities of the oligonucleotide–polymeric amine interaction. The critical length of the interacting chains is an important parameter which allows us to design sophisticated constructions containing oligonucleotide binding segments, solubilizing, protective and aiming parts. We studied interactions of (TCAG)n, n = 1–6 DNA oligonucleotides with polyethylenimine and poly(*N*-(3-((3-(dimethylamino)propyl)(methyl)amino)propyl)-*N*-methylacrylamide). The critical length for oligonucleotides in interaction with polymeric amines is 8–12 units and complexation at these length can be accompanied by “all-or-nothing” effects. New dimethylacrylamide based polymers with grafted polyamine chains were obtained and studied in complexation with DNA and RNA oligonucleotides. The most effective interaction and transfection activity into A549 cancer cells and silencing efficiency against vascular endothelial growth factor (VEGF) was found for a sample with average number of nitrogens in polyamine chain equal to 27, i.e., for a sample in which all grafted chains are longer than the critical length for polymeric amine–oligonucleotide complexation.

## 1. Introduction

Gene silencing is a promising approach to combat various diseases including cancer [1,2,3,4]. The corresponding techniques include delivery of short nucleic acids (SNA, 13–25 nucleotides) into the cell cytoplasm. Special carriers are employed to protect SNA in circulation, to facilitate SNA penetration into cell and to escape destruction of SNA in lysosomes [5]. Amine containing polymers are extensively studied in this area and the most well-known polymer in this category is polyethylenimine (PEI) [6] which became the reference in gene delivery, but, unfortunately, failed in clinical applications. To improve PEI, it may be derivatized, e.g., with a coordinative module of zinc(II)–dipicolylamine [7]. Among alternatives for PEI, the comb polymers with amine-terminated branches should be mentioned. Their main chains can consist of the sequence of methylene groups entirely, as in acrylic/methacrylic polymers [8], or biodegradable units like polyester [9]. Moreover, highly branched poly(β-amino ester)s attracted attention due to the combination of high gene transfection efficiency and low cytotoxicity [10,11,12,13]. Another way of upgrading polyplexes is involvement of a third uncharged component, e.g., multifunctional oligomers based on a lactose derivative, *N*-isopropyl acrylamide and 1-vinylimidazole and functionalized with small ligands (folate, glutathione, cysteine and arginine) endowed the formulated ternary complexes with great properties suitable for transfection [14]. High buffer capacity of PEI at pH 6–7.5 causes escape of the polymer-SNA complex from endosomes by the “proton pump” mechanism. The other requirement for SNA carriers is low cytotoxicity which is a limitation of polymeric amines. Dedicated design of the oligonucleotide carriers requires knowledge about peculiarities of SNA–polymeric amine interaction. This interaction is a typical interpolymeric reaction [15,16] and one of the important parameters of these reactions is the critical length (*L_c_*) of the interacting chains. Sub-optimal length results in weak interactions while higher *L_c_* results in relatively irreversible reactions. The importance of *L_c_* value in SNA–polymer interactions was addressed in several articles [17,18,19,20]. Synthetic polymeric amines of various length and SNA of a fixed length have been explored in several earlier reports [17,18] but the set of amine chains investigated was restricted. A recent work had investigated a set of SNA with varying chain length [19] and the minimal length of the SNA chain was determined as 14 nucleotides in reaction with virus particle of unknown chemical structure. A study of polylysine interaction with plasmid DNA [20] revealed that complexation occurred at 8 lysine units and an absence of interaction was observed with samples with 3 lysine units. The lack of information about *L_c_* in SNA–polymeric amine interactions stems from the fact that the main objective of studies in the field of gene therapy has focused on achieving complexation using SNA with definite length, usually near 20 nucleotides. However, polymeric constructs for SNA delivery often contain combinations of short amine, neutral and sometimes acidic sequences and hence information about *L_c_* values is required for the design of these constructs and understanding their properties.

Our work has two objectives with the first being estimation of *L_c_* for DNA oligonucleotides in the reaction with PEI and new polymeric amine poly(*N*-(3-((3(dimethylamino)propyl)(methyl)amino)propyl)Nmethylacrylamide) (AKX-182, Scheme 1) [21]. We have used the DNA oligonucleotides (TCAG)_n_, where n = 1–6, which contain equal amount of the each nucleotide and cannot form stable hairpins or self-dimers. The second objective of our work is to synthesize and study new polymers as gene delivery agents. These polymers originate from bioinspired polymers with grafted polyamine chains [22]. Polyamines containing 3–4 nitrogen atoms, including spermine and spermidine, play an important role in cell physiology: protein and nucleic acid synthesis, gene expression, protection from oxidative damage and etc. [23]. The so-called long-chain polyamines (LCPAs), more exactly oligomeric polyamines, have been found in bacteria [24], diatom algae [25], in the siliceous sponge [26] and in haptophyte [27]. In the case of diatoms, LCPAs are present as post-translationally grafted side chains in specific proteins namely, silaffins [28]. These LCPAs contain up to two tens of partially methylated nitrogen atoms separated by trimethylene or tetramethylene fragments [29]. Some polymeric amines [30,31] capable of interacting with oligomers of silicic acid and forming stable composite nanoparticles were considered as models of cytoplasmic silicon containing vesicles. We have elaborated [32] a method to synthesize LCPAs as an oligomeric mixture containing polyamines with 6–30 nitrogen atoms. Grafting of the LCPA chains on to poly(acrylic acid) resulted in polyampholytes which can control condensation of silicic acid giving rise to silica, the structure of which was similar to biogenic silica [33]. Neutral polymers with pendant LCPAs can interact with SNA and the obtained complexes are promising particles for gene therapy [22]. Medical applications of polymers require homogeneous macromolecules and in this work we synthesized poly(*N*,*N*-dimethylacrylamide) (PDMAAm) containing grafted LCPA chains starting from narrow polymer and LCPA fractions (Scheme 1). Influence of the length of pendant LCPA chains on the capability to interact with SNA was investigated taking into account *L_c_* values for SNA.

## 2. Materials and Methods

### 2.1. Materials

2,2’-Azobis(2-methylpropionitrile) (AIBN) (Sigma-Aldrich, St. Louis, MO, USA) was recrystallized from ethanol prior to use. Diethyl ether (Acros, Geel, Belgium) and 1,4-dioxane (Sigma-Aldrich) were distilled under sodium. Dimethylformamide (DMF) (Sigma-Aldrich) was dried with CuSO4 (30 min) and distilled at 5 mm Hg. Acryloyl chloride (Sigma-Aldrich) was distilled before polymerization. Dichloromethane (Acros, Geel, Belgium) was refluxed over phosphorus pentoxide and distilled under argon. Dimethylamine 40 wt % aqueous solution, triethylamine, 1,3-dibromopropane, *N*-hydroxysuccinimide, *N*-hydroxyphthalimide, potassium hydroxide, potassium carbonate of reagent grade (Sigma–Aldrich, Fisher, or Acros Chemicals) were used in the study. Dimethyl sulfoxide-d6 (DMSO-d6, 99.8 atom D %), deuterochloroform (99.8 atom D %), heptafluorobutyric acid (HFBA, ≥99.0% (GC)), acetonitrile (HPLC Far UV/gradient grade, Avantor Performance Materials B.V., Deventer, The Netherlands) and trifluoroacetic acid (TFA, 99 wt % purity) were purchased from Sigma-Aldrich (St. Louis, MO, USA), Fisher (Hampton, NH, USA), or Acros (Geel, Belgium) chemicals and used without further treatment. A 9.01 wt % solution of dimethylamine in 1,4-dioxane was prepared via saturation of dry 1,4-dioxane with gaseous dimethylamine. The amine concentration was determined from the resulted weight gain as well as with potentiometric titration. To obtain gaseous dimethylamine, its 40 wt % aqueous solution was added dropwise to a large excess of potassium hydroxide flakes. The evolved gas was passed through a drying column packed with KOH flakes.

*N*,*N*-bis[3-(methylamino)propyl]methylamine was prepared following the technique from our earlier work [34]. Oligo(*N*-methylazetidine) (LCPA, ZS-309 sample) was prepared from 1,3-dibromopropane and *N*,*N*-bis[3-(methylamino)propyl]methylamine according to earlier work from our group [32]. AKX-182 polymer was prepared according to the protocol described earlier [21]. PEI (M_w_ = 30,000–40,000) was obtained from SERVA Fine Biochemica (Heidelberg, Germany).

FAM 3’-tagged DNA oligonucleotides were purchased from Evrogen JSC, Russia. Small interfering RNA (si-RNA) against vascular endothelial growth factor (VEGF) was obtained from Eurofins Genomics, Louisville, KY, USA. The sequence of the sense and anti-sense strands of the fluorophore-tagged si-RNA is as follows: sense–Cy3-GGAGUACCCUGAUGAGAUC and antisense: CCUCAUGGGACUACUCUAG-Cy3.

### 2.2. Instrumentation

^1^H nuclear magnetic resonance (NMR) spectra were obtained on a DPX 400 Bruker instrument (400.13 MHz, Billerica, MA, USA) in CDCl_3_ and DMSO-d_6_. Spectra of LCPA were recorded for CDCl_3_ solutions, whereas those of poly(*N*,*N*-dimethylacrylamide)s grafted with LCPA were obtained for DMSO-d_6_ solutions of samples derivatized with trifluoroacetic acid as follows: to a sample of polymer, typically of ca. 30 mg, in a glass screw cap flat-bottom vial, 500 µL of TFA was added. The vessel was heated at about 50 °C and occasionally shaken for about 30 min to ensure complete dissolution. Then, excess TFA was removed to dryness at 50 °C with argon flow. The cooled residue was mixed with 600 µL of DMSO-d6 and left overnight at room temperature. The resultant solution was filtered through a cotton pad in a 1 mL polypropylene pipette tip directly to a NMR ampoule.

Mass spectrometric analysis was performed on an Agilent 6210 TOF LC/MS System. The samples were dissolved in acetonitrile. Water and acetonitrile with 0.1% (*v*/*v*) HFBA were used as eluting solvents A and B, respectively (Solvent A–90%, B–10%). The flow rate of the mobile phase was set at 0.2 mL/min, while the injection volume of sample solution was 20 μL. The conditions for TOF MS were as follows: the mass range was *m*/*z* 100–1000 and scan time was 1 s with an interscan delay of 0.1 s; mass spectra were recorded under ESI+, V mode, centroid, normal dynamic range, capillary voltage 3500 V, desolvation temp 350 °C, nitrogen flow 5 L/min.

The molecular masses of the new polymers were estimated via size-exclusion chromatography (SEC) using a Milichrom A02 chromatograph (JSC Econova, Novosibirsk, Russia) with 2 mm × 75 mm column filled with SRT SEC-100 5 μm phase (Sepax Technologies, Inc., Newark, NJ, USA), operated at 35 °C using phosphate buffer solution 0.15 M, pH 6.86. The flow rate of the mobile phase was set at 0.03 mL∙min−1 (pressure 100 psi), whereas the injection volume for 1 g∙L^−1^ of the sample solution was 1 μL. Fractionated samples of poly(vinyl formamide) [35] were applied as standards (M_w_/M_n_ < 1.3).

Zeta-potential (ζ) and hydrodynamic radius (R_h_) were measured at 25 °C with a Zetasizer Nano-ZS ZEN3600 (Malvern Instruments Ltd., Worcestershire, UK), equipped with a 4mW He-Ne laser operating at λ_o_ = 633 nm. Measurements were performed at θ = 173° in zeta-potential-DTS1060 folded capillary cells. Calculation of ζ and R_h_ values was performed using software supplied with the Zetasizer.

Transmission electron microscopy (TEM) of the polymer-DNA complexes was performed using a LEO 906E instrument on solutions diluted tenfold just before freezing. The solutions were placed on formvar film coated copper grids and air dried.

Atomic force microscopy (AFM) was performed using Scanning Probe Microscope CMM-2000 (PROTON-MIET, ZAVOD, JSC, Zelenograd, Russia) operated in contact mode in air at room temperature using silicon probes (nominal probe curvature radius of 10 nm). Height mode images (512 × 512 pixels) were collected with a scan speed between 1 and 2 Hz. The samples were placed on mica slips, water was removed with a filter paper after 30 min and slips were air dried. The software package Gwyddion was used for AFM image processing.

### 2.3. Synthesis of Poly(N,N-dimethylacrylamide) with Grafted LCPA Chains (ZS-371-n)

#### 2.3.1. Synthesis of Poly(acryloyl chloride) (PAC)

PAC was synthesized similar to the protocol described earlier [36] by polymerization of acryloyl chloride (6.336 g, 0.070 mol) in 25 mL of dioxane with the addition of 0.1267 g AIBN in argon atmosphere at 60 °C for 48 h. PAC was precipitated from the reaction solution using 138 mL of cyclohexane. The supernatant was decanted and the residue was dissolved in DMF (Solution A, see below) for further use. With the objective to estimate yield and polymerization degree of the PAC, the reaction mixture was poured into water (50 mL) and dialyzed against water. After freeze drying, poly(acrylic acid) was obtained with 90% yield. According to viscometry data, [37] the polymerization degree of the poly(acrylic acid) and, correspondingly of PAC, was found to be 220.

#### 2.3.2. Preparation of Poly(*N*,*N*-dimethylacrylamide-co-*N*-acryloxysuccinimide) (ZS-358)



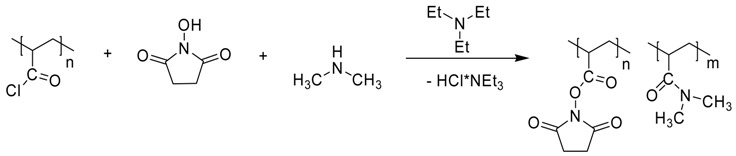



Three initial solutions namely Solution A (The prepared PAC in 104.4 g of DMF), Solution B (*N*-hydroxysuccinimide (1.61 g, 14.0 mmol) and triethylamine (1.70 g, 16.8 mmol) dissolved in 11 g of DMF), and Solution C (Triethylamine (7.22 g, 71.4 mmole), 10.1 wt. % dimethylamine solution in 1,4-dioxane (26.55 g, corresponds to 59.48 mmol of HN(CH3)2) and 22.9 g of DMF) were combined.

Solution B was added to magnetically stirred solution A (cooling on an ice bath) for eight minutes. The mixture was kept stirred with cooling for 31 min followed by dropwise addition of solution C for 30 min. Stirring was continued with cooling for 30 min and then at room temperature for two hours. The reaction vessel was left in a refrigerator for about 15 h. The white precipitate was filtered off using a filter funnel with sintered glass disc. The light yellowish filtrate was concentrated to a volume of 10–15 mL under vacuum at room temperature. Addition of 110 mL of toluene precipitated a yellow sticky mass which was washed with toluene (50 mL × 1), THF (25 mL × 2) and dried using an oil vacuum pump for four hours to yield 7.496 g of ZS-358.

#### 2.3.3. Fractional Precipitation of ZS-358

A solution of ZS-358 (6.434 g) in 26 mL of CH_2_Cl_2_ was filtered through a glass disc (10–15 μm) and diluted to a total volume of 215 mL with CH_2_Cl_2_. The first fraction (1F) was precipitated with careful addition of 19.64 g of n-hexane to the stirred solution. The supernatant was decanted in about 90 min for the successive precipitations (Table 1).

#### 2.3.4. Preparative Exclusion Chromatography (SEC) Fractionation of ZS-309

The fractionation was performed on a glass jacketed chromatography column (0.7 cm × 85 cm) packed with Sephadex G-25 (coarse, 100–300 μm). A solution of 100 mg of ZS-309 in 200 μL of 0.1 M HCl was loaded on the wet sorbent surface followed by gravity elution at 35 °C with a 25 mM acetate buffer solution containing 0.2 M NaCl. Collected fractions were analyzed by silica gel TLC (CH_2_Cl_2_:CH_3_OH: 25% aq. NH_3_ = 2:2:1). The spots were visualized in red color by Dragendorff’s reagent [38]. Chosen fractions were combined and analyzed with LC-MS (Appendix A) with the objective to estimate effectiveness of the fractionation. Then, the solutions were evaporated to dryness under vacuum, mixed with a solution of K_2_CO_3_ (50%) in water and extracted with CH_2_Cl_2_. The extracts were dried over anhydrous potassium carbonate, evaporated and finally kept under vacuum of an oil pump for two days. The obtained LCPA fractions contain 27.6 (ZS-309-1), 16.3 (ZS-309-2), 13.4 (ZS-309-3) nitrogen atoms in molecule as determined with ^1^H NMR (Appendix A) with the use signals at 2.40 ppm (terminal methyl groups) and 2.16 ppm (methyl groups in the polyamine chain).

#### 2.3.5. Grafting of Oligo(*N*-methylazetidine) onto Poly(*N*,*N*-dimethylacrylamide-co-*N*- acryloxysuccinimide) (ZS-371-n)



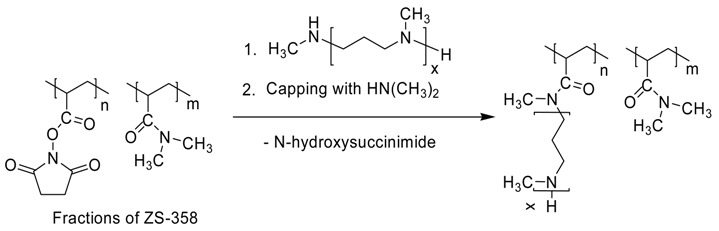



Preparation of ZS-371-1. ZS-309-3 (0.06602 g), a fraction of oligo(*N*-methylazetidine), was dissolved in a mixture of 0.70 g of DMF and 1.02 g of 10.8% HN(CH_3_)_2_ in 1,4-dioxane (1-st portion) followed by addition of 0.24958 g of 3F (a ZS-358 fraction) solution in 1.83 g of DMF. The solution was purged with argon, sealed in a glass vessel and kept at 60 °C for 24 h. After that, the 2-nd portion of the dimethylamine solution (0.38 g) was added and heating continued for additional 24 h. Then, the mixture was rotary evaporated and further kept under vacuum of an oil pump at room temperature for 3 h. The sticky residue was thoroughly triturated with diethyl ether multiple times, the ether solutions discarded and the product reprecipitated three times from methylene chloride/methanol to ether. After vacuum drying, the polymer was dissolved in deionized water, filtered through a 0.45 μm cellulose acetate membrane and freeze-dried to give 0.222 g of ZS-371-1.

The other polymers ZS-371-(2-7) were prepared similarly, see the Table 2.

The content of grafted units was calculated from ^1^H NMR spectra (Appendix A) of TFA-derivatized samples dissolved in DMSO-d_6_ [39]. The integral intensities of the 1H NMR signal at 10.4 ppm (protons at tertiary nitrogen atoms) were compared with the integral intensities of the all C-H protons within 0.8–4.6 ppm except for those of DMSO, keeping in mind the average length of the grafted oligoamines.

### 2.4. Study of Polymer–Oligonucleotide Interactions and In Vitro Activity of the Polyplexes

#### 2.4.1. Interaction with DNA Oligonucleotides

The interaction between DNA oligonucleotides and polymers was investigated by electrophoresis on agarose gel. Complexes were prepared by mixing solutions of the polymer and oligonucleotide. The samples were incubated at room temperature for 30 min and placed into the wells of the 1% agarose gel. Free oligonucleotide, as a control, was also loaded onto the gel. The gel running buffer was 40 mM Tris acetate (adjusted to pH 7.4) and 1 mM EDTA. A glycerol gel loading buffer was used (0.5% sodium dodecyl sulfate, 0.1 M EDTA (pH = 8), and 50% glycerol for 10× reagent). The gel was run at 90 V and the fluorescein-tagged oligonucleotide was visualized on a UV transilluminator.

#### 2.4.2. RiboGreen Assay

Polyplexes were prepared in the ratio of 4:1 and diluted to 100 µL with deionized water in a 96–well plate. 100 µL of 1:1000 diluted ribogreen dye (Invitrogen, Carlsbad, CA, USA) was added to each well and the fluorescence intensity was measured using a multimode reader (Synergy H1, Biotek, Winooski, VT, USA) following incubation for 5 min in dark. The sample was excited at 490 nm and the emission was measured at 525 nm.

#### 2.4.3. Study of the Polymer and Polyplex Toxicity

Toxicity of the polymer and polyplex was evaluated using (3-(4, 5-dimethylthiazol-2-yl)-5-(3-carboxymethoxyphenyl)-2-(4-sulfophenyl)-2H-tetrazolium) MTS assay (Cell Titer 96 Aqueous one solution, Promega, Madison, WI, USA). Four thousand A549 cells per well were seeded in a 96-well plate and incubated at 37 °C in 5 % CO_2_. After the cells achieved confluency, the medium was removed and washed with PBS (pH 7.4) to remove the non-adherent cells. Polyplexes were prepared by mixing 10 µM siRNA (Eurofins, Louisville, KY, USA) with 4 mg/mL polymer solution in 1:4 ratio. Then, 5 µL of the polyplex or 2 mg/mL polymer solution was mixed with 100 µL of serum-free media and added to the cells. After 4 h, the medium was replaced with the fresh medium and incubated for 24 h or 48 h. MTS reagent (Promega, Madison, WI, USA, 10 µL) and 100 µL of serum-free media was added to each sample well and incubated at 37 °C for 2 h. The reaction was stopped by addition of 25 µL of 10% sodium dodecyl sulfate (SDS) solution. The absorbance was measured at 490 nm using multimode reader (Synergy H1, Biotek, Winooski, VT, USA).

#### 2.4.4. Study of the Polyplex Internalization

These studies were carried out with a polymer concentration of 4 mg/mL and a siRNA concentration of 10 µM with a polymer to siRNA ratio of 4:1. The internalisation of the polyplex in A549 cells was evaluated using fluorescent siRNA (Eurofins) that is excited at 538 nm and emits at 640 nm. A549 cells were cultured in DMEM (GIBCO, Thermo Fisher Scientific, Waltham, MA, USA) on a cover slip in a 6-well plate with a seeding density 10^5^ cells/well. After the cells attained confluency, the medium was removed and washed with phosphate buffered saline (PBS, pH 7.4) to remove the non-adherent cells. Then, 5 µL of polyplexes were added to 200 µL of serum-free media and incubated for specific time points. The medium was then replaced with fresh medium containing fetal bovine serum (GIBCO, Thermo Fisher Scientific, Waltham, MA, USA). The cells were stained with Hoechst 33342 (Invitrogen, Carlsbad, CA, USA) and the images were captured using laser scanning confocal microscopy (FV1000, Olympus, Tokyo, Japan).

#### 2.4.5. Western Blot Study of VEGF Silencing

For performing Western blot, total protein was isolated from the A549 cells using cell lysis buffer that contains a cocktail of RIPA buffer, PMSF, protease and protease inhibitors (Cell Signalling, Danvers, MA, USA). The isolated protein was quantified using Lowry’s method. An aliquot containing 50 mg protein was subjected to electrophoresis using 12% SDS-PAGE (sodium dodecyl sulfate-polyacrylamide gel electrophoresis). The protein bands were transferred to PVDF membrane at 4 °C for 1 h in the presence of transfer buffer containing tris-glycine. Non-specific binding was eliminated using a blocking agent of 5% skimmed milk in Tris-buffered saline containing 0.1% Tween-20. The membrane was incubated overnight with primary antibody (VEGF antibody, dilution 1:500, Santa Cruz, CA, USA) at 4 °C. The blots were then washed with tris-glycine buffer followed by incubation with anti-mouse horseradish peroxidase-conjugated secondary antibody (dilution 1:5000, Cell Signalling, Danvers, MA, USA) for 1 h at room temperature. The protein bands were visualized using Tetramethyl benzidine/hydrogen peroxide (TMB/H_2_O_2_, Bio-Rad, Hercules, CA, USA) reagent following manufacturer’s protocol and the images were captured using gel documentation system (Chemidoc, Bio-Rad, Hercules, CA, USA). The housekeeping gene beta actin was visualized after stripping the membrane following which the membrane was reblocked and re-incubated with anti-beta actin primary antibody (Cell Signalling, Danvers, MA, USA). Quantity-one software was used for analysis of the images and the band intensities were calculated. After background normalization, beta-actin band intensity was used to normalize the band intensity of the VEGF protein band.

## 3. Results and Discussion

### 3.1. Determination of Critical Length of DNA Sequence in Reaction with Polymeric Amines

The influence of length of single-strand DNA oligonucleotide on the ability to interact with PEI and AKX-182 was studied at an N/P ratio (ratio of amine groups to nucleotides) equal to 20. The high excess of polymeric amine minimizes stacking and other possible side effects. Gel electrophoresis data (Figure 1) show formation of slightly positive complex with 12–24-mer DNA for PEI and 16–24 mer DNA in the case of AKX-182. Decreasing the DNA length to 8–12 units results in a diffuse spot which corresponds to some negative-charged complexes. 4-mer DNA does not interact with PEI and AKX-182. The formation of negative-charged products is unexpected, poor interaction of short oligonucleotides with polymeric amines could result in the presence of free DNA but the observed diffuse spots are far from free DNA position. Here, we see two kinds of DNA-containing particles: neutral (or slightly positive) and negative charged. These complexes were observed at various N/P ratios and DNA length (Table 3). Increasing the DNA length and N/P ratio results in disappearance of the negatively charged complex. Nucleic acids can interact with amines by the means of ionic interactions through phosphate groups or by hydrogen bonds through nucleobases [40,41] and references in this review. The latter mechanism suggests that the negatively charged free phosphate groups compensates the positive charge of the free amine units. We can hypothesize formation of negatively charged coordinated regions surrounded by positively charged parts of the polymer (Scheme 2a). Increase in the polymeric amine content results in the presence of a positively charged complex only. The formation of only negative particles with short DNA chains is possibly due to the “all-or-nothing” scheme (Scheme 2b) which is often realized with weak associative interactions [42,43]. Experiments with double stranded DNA (Figure 2) show similar behavior of the DNA-polymer complexes.

Thus, 8–12 nucleotides represent a critical length of DNA chains at which interaction with polymeric amines proceeds under high excess of the amine and is accompanied by “all-or-nothing” effects.

### 3.2. Synthesis of Poly(N,N-dimethylacrylamide) with Grafted LCPA Chains

The desired PDMAAm with grafted LCPA chains was synthesized starting from poly(acryloyl chloride) which was converted into PDMAAm containing activated acrylic ester (oxysuccinimide) units. This copolymer was fractionated by precipitation from CH_2_Cl_2_ with n-hexane. The mixture of LCPA oligomers was fractionated with flash-SEC and three fractions containing oligomers with average 13, 16 and 27 nitrogen atoms were obtained. These oligomers as well as anon-fractioned sample were involved in the reaction with activated polymer, giving rise to the target polymers (Scheme 3, Table 2). The grafting degree of the LCPA chains is 1.8–3.4, M_w_/M_n_ ratio is 1.16–1.24 for polymers with fractionated LCPA chains which is significantly lower than the values for polymers based on poly(acryloyl chloride) without fractionation (1.37–1.58 [21]).

### 3.3. Interaction of LCPA Containing Polymers with Oligonucleotides

Complexation between PDMAAm with grafted LCPA and oligonucleotides was studied with gel electrophoresis (Figure 3). The AKX-371-5 sample only interacts with 16–24-mer DNA N/P = 20. The other polymers yield complexes at N/P > 40. This behaviour corresponds to the data with PEI and AKX-182 samples: AKX-371-5 contains long (>16 units) side polyamine chains and all these chains exceed the critical length for interaction with DNA. AKX-371-1 (LCPA chain length–13.4) is not active in the complexation because the LCPA chain is not sufficient for the reaction with DNA and stacking structures from several LCPA chains are not possible due to the low grafting degree. AKX-371-2 and AKX-371-3 contain a certain fraction of LCPA longer than 20 units and an increase in the polymer concentration results in approximately full complexation at N/P = 150.

Interaction of the polymers with 19-mer double stranded si-RNA against vascular endothelial growth factor (VEGF) was studied with RiboGreen assay [44] which allows us to measure the concentration of free, non-complexed RNA (Table 4). The most stable complexes were obtained with the ZS-371-5 polymer and the complexation proceeds to a large degree at N/P ratio 20, which corresponds to formation of partially negatively charged complexes.

DLS data (Figure 4) show the presence of ZS-371-1, ZS-371-2 and ZS-371-3 macromolecules in a non-aggregated state, R_h_ = 3–4 nm is close to the size of other polymers based on PAC [45]. Interaction with nucleic acid results in the formation of aggregates, especially in the case of ZS-371-2 and ZS-371-3. The ZS-371-1 polymer forms smaller aggregates probably because short LCPA chains cannot compensate the negative charge of siRNA, which is confirmed with ζ-potential data. ZS-371-5 exists in the form of 50 nm aggregates and complexation with siRNA does not influence the particle size. We can hypothesize that siRNA chains interact with LCPA chains of comparable length without changing the aggregate structure. TEM (Figure 5) and AFM (Figure 6) data qualitatively agree with DLS data. The larger size values obtained with DLS are explainable with swelling of the macromolecules and association with hydrated counterions. In any case, complexes of the LCPA containing polymers with oligonucleotides exist as 100–200 nm particles (Figure 4) which is appropriate for internalization into living cells [46,47,48].

### 3.4. In Vitro Study of Transfection Activity of si-RNA Polyplexes Based on LCPA Containing Polymers

The new polymeric amines were studied as transfection agents with the use of 19-mer si-RNA which can silence vascular endothelial growth factor (VEGF) [49]. Viability (Figure 7) and internalization assays (Figure 8) were performed to estimate the transfection potential of the new polymers.

The free siRNA does not show significant reduction in viability after 24 h as well as after 48 h. The free ZS-371-1 shows a slight but insignificant decrease in the viability of the A-549 cells after 48 h but its complex with anti-VEGF siRNA does not show any toxicity after both time points. This suggests that the surface charges on the uncomplexed polymer may interfere with the viability of the proliferating cells. Complexation with the siRNA neutralizes the surface charges that are manifested in the absence of any marked decrease in the cell viability after 48 h. The ZS-371-2 polymer does not alter the viability at both time points, indicating its lack of toxicity. However, its polyplex with siRNA shows a slight reduction in the viability after 48 h. This suggests an effect of the VEGF silencing on different signalling pathways in the polyplex treated cells. In comparison, the ZS-371-3 did not alter the cell viability after 24 h or 48 h both in the uncomplexed as well as complexed forms. This may arise due to poor internalization into the cells or due to poor complexation or extremely high affinity complexation that restricts the release of the siRNA or due to poor ability to escape from the endosome. However, the internalization studies (Figure 8a) show that the ZS-371-3 polyplex localizes rapidly within 4 h in the cells, thereby ruling out the possibility of poor internalization. Further, ribogreen assay reveals that the complexation efficiency is about 92% and hence poor siRNA could not explain the observed cell viability. Therefore, it could be possible that the siRNA release or endosomal escape may be the factors that could have influenced the lack of toxicity for this formulation. The viability of cells treated with ZS-371-5 shows a small but insignificant reduction in the viability after 48 h but its complex with the anti-VEGF siRNA shows a significant decrease in the cell viability after 48 h, indicating the effect of VEGF silencing in the cells. Several reports [50,51] have indicated that VEGF silencing can mediate cell death and decreased proliferation by modulating the PI3K/Akt and Notch signalling pathways, respectively. Internalization studies show (Figure 8b) significant internalization of si-RNA after 6h. siRNA localizes mostly in cytoplasm (not in nucleus) which is necessary for effective silencing. The ribogreen assay revealed that ZS-371-5 shows the maximum complexation efficiency followed by the ZS-371-2 which is explainable with the length of the grafted LCPA chains. This could be one of the factors that could have influenced the cell viability assay. Additional factors such as endosomal escape and release of siRNA, which could also have influenced the viability, need further validation.

The VEGF silencing efficiency of the polyplexes was evaluated using Western blot and the results are presented in Figure 9. Though all polyplexes show a reduction in the VEGF band intensity when compared with the untreated control cells, the maximum reduction in the VEGF band intensity was observed for ZS-371-5 polyplex while the least reduction was observed for ZS-371-1. Upon normalizing the band intensities with beta actin, the maximum silencing efficiency was obtained for ZS-371-5 and ZS-371-3 while ZS-371-1 and ZS-371-2 showed lesser silencing efficiency. Thus, the loss in viability observed in A549 lung cancer cells treated with the polyplex ZS-371-5 may be attributed to a direct consequence of its VEGF silencing ability.

## 4. Conclusions

We have found that the critical length for oligonucleotides in interaction with polymeric amines is 8–12 units and complexation at these lengths can be accompanied by “all-or-nothing” effects. New polymers with grafted polyamine chains were obtained and studied in complexation with DNA and RNA oligonucleotides. The most effective interaction, transfection activity and VEGF silencing efficiency was found for a sample with an average number of nitrogens in polyamine chain equal to 27, i.e., for a sample in which all grafted chains are longer the critical length for polymeric amine–oligonucleotide complexation.

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
