# Peer review of "Design of Oligonucleotide Carriers: Importance of Polyamine Chain Length"

_polymers, 2018, doi:10.3390/polym10121297_

Round 1

Reviewer 1 Report

In this report, the authors reported the importance of polyamine chain length on the gene transfection performance of oligonucleotides. They synthesized a series of polycations and systematically studied the interactions with oligonucleotides and identified the critical length of the interacting chains. This work is of great significance to the field of gene therapy because it provides new insights for researchers to design gene delivery systems. The data support the conclusions very well. I recommend to accept this manuscript for publication after addressing the following issues:

In the introduction part of the manuscript, the authors should give some introduction on other polycations that have been widely used for gene therapy and cite the relevant papers (e.g., Science Advances, 2016, 2, e1600102; Journal of the American Chemical Society, 2017, 139, 5102; Journal of Controlled Release, 2016, 244, 336; New Journal of Chemistry, 2016, 40, 9806; Biomaterials Science, 2016, 4, 522,; Biomaterials Science, 2016, 4, 93; ACS Biomaterials Science & Engineering, 2016, 3, 1283; Biomacromolecules, 2016, 17, 3640; etc.). This is very important because PEI has been demonstrated to be not the best candidate for gene therapy although it has been widely used. In contrast, many newly developed polymers are more viable for potential clinical applications.

The authors should provide more information on the gene transfection efficiency.

The authors used gel electrophoresis and Ribogreen assays to study the interaction between polycations and SNA, did they measure the size and zeta potential of the formulated complexes?

Author Response

Dear Sir/Madam,

Many thanks for your benevolent attention on our manuscript! We have revised the text according to your comments.

Sincerely yours,

Vadim Annenkov

Replies on the comments:

In the introduction part of the manuscript, the authors should give some introduction on other polycations that have been widely used for gene therapy and cite the relevant papers (e.g., Science Advances, 2016, 2, e1600102; Journal of the American Chemical Society, 2017, 139, 5102; Journal of Controlled Release, 2016, 244, 336; New Journal of Chemistry, 2016, 40, 9806; Biomaterials Science, 2016, 4, 522,; Biomaterials Science, 2016, 4, 93; ACS Biomaterials Science & Engineering, 2016, 3, 1283; Biomacromolecules, 2016, 17, 3640; etc.). This is very important because PEI has been demonstrated to be not the best candidate for gene therapy although it has been widely used. In contrast, many newly developed polymers are more viable for potential clinical applications.

Reply: We have added the recommended references (p. 1).

The authors should provide more information on the gene transfection efficiency.

Reply: We have estimated the gene transfection efficiency using Western blot study of VEGF silencing. The results confirm the high activity of ZS-371-5 sample. A text was added into experimental (p. 7), Figure 10 and discussion (p. 15).

The authors used gel electrophoresis and Ribogreen assays to study the interaction between polycations and SNA, did they measure the size and zeta potential of the formulated complexes?

Reply: We have done these experiments (p. 4 – experiment, Figure 5 – data and p. 11 – discussion).

Reviewer 2 Report

The authors studied the critical length of polymer chain effect on the efficacy of binding with SNA. However, the manuscript at the current stage should not be accepted for publication. Revision is recommended.

1. The major concern: please show the gene and protein data about siRNA efficacy after they bind with polymers. I strongly suggest to investigate the VEGF gene expression and VEGF protein secretion data using PCR and ELISA after the cells interacted with the polymer carrying anti-VEGF siRNA.

2. The quality of the paper should be improved:

(1) no statistical analysis seen in the viability data.  Please show standard deviations for all quantitative data.

(2) The gel electrophoresis image quality should be improved. Some images are not clear.

(3) Authors don't need to show two similar AFM images for one sample. In addition, AFM is not the best tool for characterizing the particle formation . Please use TEM and dynamic light scattering.

Author Response

Dear Sir/Madam,

Many thanks for your benevolent attention on our manuscript! We have revised the text according to your comments.

Sincerely yours,

Vadim Annenkov

Replies on the comments:

1. The major concern: please show the gene and protein data about siRNA efficacy after they bind with polymers. I strongly suggest to investigate the VEGF gene expression and VEGF protein secretion data using PCR and ELISA after the cells interacted with the polymer carrying anti-VEGF siRNA.

Reply: We have estimated the gene transfection efficiency using Western blot study of VEGF silencing. The results confirm the high activity of ZS-371-5 sample. A text was added into experimental (p. 7), Figure 10 and discussion (p. 15).

2. The quality of the paper should be improved:

(1) no statistical analysis seen in the viability data.  Please show standard deviations for all quantitative data.

Reply: The corresponding data present on Figure 8.

(2) The gel electrophoresis image quality should be improved. Some images are not clear.

Reply: The image quality was improved (Figures 1 and 4).

(3) Authors don't need to show two similar AFM images for one sample. In addition, AFM is not the best tool for characterizing the particle formation . Please use TEM and dynamic light scattering.

Reply: We do not show two similar AFM images for one sample, the images differ in some conditions (single/double strand). We have done all recommended experiments (p. 4 – experiment, Figures 5 and 6 – data and p. 11 – discussion).

Round 2

Reviewer 2 Report

Thanks for the authors’s efforts for revising the manuscript. The manuscript looks much better than the oringal one. The only thing that may need the author to change: please add statistical analysis for the quantitative data in the figures.

Author Response

Dear Sir/Madam,

Many thanks for your benevolent attention on our manuscript! We have revised the text according to the  comment.

Sincerely yours,

Vadim Annenkov

Replies on the comment:

The only thing that may need the author to change: please add statistical analysis for the quantitative data in the figures.

Reply: As we understand, according to the previous comments, the Referee has in mind Figures 8 and 10. We had the error bars on these figures and now we add text like "Data represent mean ± SD (*p < 0.05, n = 5)". As far as I know, this is an usual way to present such quantitative data, e.g. [Liu, S.; Gao, Y.; Sigen, A.; Zhou, D.; Greiser, U.; Guo, T.; Guo, R.; Wang, W. Biodegradable highly branched poly(β-amino ester)s for targeted cancer cell gene transfection. ACS Biomater. Sci. Eng. 2017, 3, 1283–1286, DOI: 10.1021/acsbiomaterials.6b00503].